# Comparative Study on Fuel Gas Supply Systems for LNG Bunkering Using Carbon Dioxide and Glycol Water

**Jungho Choi *** and **Eun-Young Park**

Department of Naval Architecture and Offshore Engineering, Dong-A University, Busan 49135, Korea; pyy0924@naver.com
* Correspondence: tamuchoi@dau.ac.kr; Tel.: +82-51-200-7938

**Abstract:** Liquified natural gas (LNG) fuel has received significant attention as an affordable and highly efficient fuel option due to strengthened regulations on the sulfur content of bunker oil put in place by the International Maritime Organization. The main component of the LNG fuel gas supply system (FGSS) is the heat exchanger that provides adequate gas temperatures and pressures required by the engine, which also has a large machinery volume compared with other equipment. Due to the volume limitation, most FGSS have been applied to new shipbuilding only. To reduce the volume of the FGSS, $CO_2$ was considered to serve as the replacement heat medium for conventionally used glycol water during LNG gasification. The specific power consumption (SPC) in the $CO_2$ and glycol water system was optimized using the Aspen HYSYS thermodynamic modeler toward adjusting the temperature and pressure, and the resulting sizes were compared. This study demonstrated that the $CO_2$ heat medium resulted in a 14% improvement in efficiency and a 7% reduction in heat exchanger size concluding that it was the most advantageous heat medium for the LNG regasification.

**Keywords:** fuel gas supply system; heat exchanger; $CO_2$; LNG bunkering

## 1. Introduction

The 70th Session of the Marine Environment Protection Committee (MEPC) was held in 2016, where the International Maritime Organization (IMO) decided to strengthen regulations on the sulfur content of bunker oil by reducing them from 3.5% to 0.5% starting in 2020. At present, to adhere to the strengthened ship emission regulations, there are three alternative options, which include: low sulfur fuel oil (LSFO), selective catalytic reduction (SCR), and liquefied natural gas (LNG) used as a fuel to operate LNG-fueled ships. LNG-fueled ships generate approximately zero sulfur oxide (SOx) and particulate matter (PM) emissions and their affordable fuel cost and high efficiency ensure economic feasibility. Daewoo Shipbuilding & Marine Engineering constructed a downsized fuel gas supply system (FGSS) that compresses NG in liquid state to reduce the power consumption of the pump [1,2].

These ship types require an additional FGSS that provides adequate gas temperatures and pressures required by the engine to use the stored LNG as a fuel. The heat exchanger used in these systems results in a larger machinery volume and, as such, it is difficult to install the LNG FGSS after the ship remodeling. Therefore, potential solutions to reduce the volume of the heat exchanger have attracted significant research interest. The optimal design of a heat exchanger would maximize heat transfer while minimizing the production cost, operation cost, and volume. However, there is a limit to the size reduction of the heat exchanger as it is directly related to the heat transfer capacity of the heat exchanger.

With respect to the research and development (R&D) trend for LNG FGSSs, the primary feature of the LNG FGSS Linde unit [3] comprises of two 400 m$^3$ insulated vacuum tanks installed vertically

beneath the deck. This design is intended to ensure the safe and efficient storage of LNG in a variety of ships. Previous studies on LNG gasification have mainly focused on system improvements related to the refrigerant type, as described herein. The VALMAX Technology [4] LNG Regasification System uses glycol water as a heat medium for the LNG vaporizer, as well as indirect heating to solve problems associated with corrosion due to the direct heating of seawater. The MAN Diesel & Turbo MEGI pump vaporizer unit uses glycol water as a heat source. Samsung Heavy Industries uses glycol water and waste heat from the marine engine for the vaporizer heat to achieve a high degree of efficiency. Sun et al. [5] proposed a new Rankine power cycle that uses three hydrocarbon mixtures as the working fluid to utilize the cold energy of LNG. Lee et al. [6] used glycol water as a heat medium when constructing a regasification system and analyzed the system according to the mixing ratio between glycol water and water. This study also estimated the temperature of the natural gas supplied to the dual fuel (DF) engine and the temperature of the glycol water supplied to the heat exchanger. Garcia et al. [7] used argon and methane as working fluids, taking advantage of energy loss and efficiency using the Rankine cycle. Lee [8] optimized process parameters in the cascade Rankine cycle using C2 and C3 as the working fluid and thermodynamic modeling. Mehrpooya et al. [9] used a mixture of $NH_3$ and $H_2O$ as a refrigerant to simultaneously reduce both the energy and cost, thereby enhancing component performance.

There have also been previous studies related to modifications of the system process conditions. Shi and Che [10] used a refrigerant that was a mixture of $NH_3$ and $H_2O$ to produce a model using the Rankine and LNG generator cycles, and comparatively analyzed the energy efficiency while changing the inlet and outlet pressures at the turbine. Lu and Wang [11] proposed a stepped power cycle using ammonia water as the working fluid to recover LNG cryogenic energy. Kim et al. [12] analyzed the LNG regasification system for the basic process design condition through indirect heating or direct heating of steam or seawater. Park et al. [13] carried out an alternative to minimize thermal and chemical exergy losses, which has been shown to improve overall rational emission efficiency of the process. Liu and Guo [14] compared and analyzed the cycle-based method to improve the recovery efficiency of LNG cool energy. Gomez et al. [15] evaluated the LNG exergy on a thermal basis based on the cycle. Lee and Choi [16] thermodynamically evaluated an integrated heat recovery system that simultaneously recovered LNG cold energy and diesel generator exhaust heat. Li et al. [17] assessed thermal performance by analyzing the LNG cold warehouse under different pressures.

DaeChul et al. [18] examined the characteristics of LNG gasification according to changes in LNG inflow and glycol water flow to derive the optimal operating conditions for gas discharge in LNG ships. Tomkow and Cholewinski [19] focused on enhancing energy efficiency by using ethanol as a refrigerant and coupling it with the organic Rankine cycle to prevent potential energy loss. Tan et al. [20] applied an ejector to the conventional LNG re-liquefaction system to enhance energy efficiency and reduce power requirements using two cascade dual mixed refrigerant cycles. Bao et al. [21] observed changes in the cost index, and the heat transfer area (UA), as a function of changes in the condensation temperature of the heat exchanger in the system, consequently achieving energy recovery based on the condensation temperature of the heat exchanger outlet. Lee and Mitsos [22] investigated changes in system power as a function of temperature changes in the heat source and confirmed a positive correlation between the heat efficiency of the system and the temperature of the heat source. Lim and Lee [23] used toluene and benzene as working fluids and comparatively analyzed the cycle performance difference as a function of changes in overheating at the turbine inlet. Choi et al. [24] applied five types of working fluids to the system and analyzed the cycle performance as a function of changes in the vaporization and condensation temperatures of the system.

There is currently an interest from the relevant industries in the minimization and efficiency of the gasification system; however, most previous studies have focused on improving efficiency by modifying the system composition, refrigerant selection, and operating point. In contrast, this study aims to reduce the size of the heat exchanger since it has the largest footprint in the FGSS while increasing the efficiency of the heat transfer process. Thus, the objective of this study is to minimize

the overall device size by increasing the efficiency of the heat transfer process using the latent heat of $CO_2$ vaporization since $CO_2$ is a heat source with a phase change.

## 2. Analytical Methods

### 2.1. Comparing Alternative Heat Sources to Seawater

The characteristics of alternative heat media as a replacement for seawater within the FGSS are outlined in Table 1. Although glycol water is frequently used as a heat medium by shipbuilders due to its high specific heat and wide operating temperature, there is a distinct disadvantage when using this media due to corrosion and the risk of toxicity. In this study, we used $CO_2$ since it is a low cost, non-toxic, and non-flammable medium; however, caution should be used with respect to the pressure at the triple point during system design. Moreover, $CO_2$ suffers from another disadvantage, in that it is a corrosive medium when contaminated by small amounts water. If the pressure increases, or the temperature decreases at this point, it will result in $CO_2$ solidification, preventing gas flow.

**Table 1.** Characteristics of the refrigerant media.

|                | Advantage          | Disadvantage | Operating Temperature (°C) | Risk                 |
| -------------- | ------------------ | ------------ | -------------------------- | -------------------- |
| **Glycol Water** | High Specific heat | Corrosion    | (−30) to 106               | Toxicity             |
| **Pure $CO_2$**  | Affordable         | Icing        | (−78.5) to 31.4            | PEL [1]: 5000 ppm    |

[1] PEL: permissible exposure limit.

The correlation between $CO_2$, the heat source used in this study, and glycol water, commonly used as a heat source in the gasification of cryogenic LNG, is shown in Figure 1a and is represented by Equation (1):

$$Q_{LNG} = UA \cdot LMTD_{@glycol\ water,\ CO_2} \tag{1}$$

The area of the T-Q diagram in Figure 1a corresponds to the log mean temperature difference (LMTD) term in Equation (1). Compared with conventional glycol water, $CO_2$ is characterized by a significantly lower area due to its latent heat of vaporization. This indicates a high UA (i.e., product of the overall heat transfer coefficient multiplies heat transfer area) in the heat exchanger for heat exchange with $CO_2$ during the gasification of the same LNG flow. Although an increase in UA in a heat exchanger implies an increase in the heat transfer area, this is likely influenced by the overall heat transfer coefficient (U).

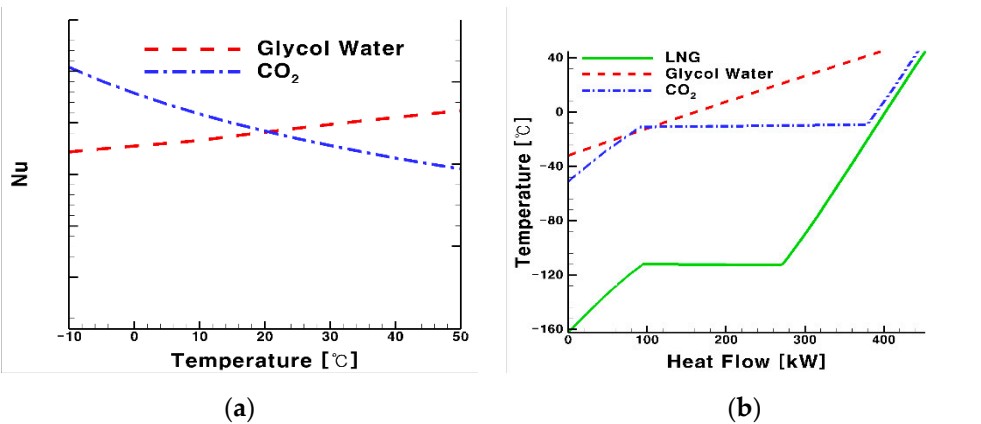

(a)                                                                    (b)

**Figure 1.** Heat transfer performance based on the refrigerant and heat process: (**a**) T-Q diagram classified by heat source; (**b**) Nusselt number (Nu) as a function of temperature for glycol water and $CO_2$. LNG: liquefied natural gas.

Heat transfer performance is represented using the Nusselt number (Nu) value for $CO_2$ and glycol water as a function of temperature, whose trends are shown in Figure 1b. In the low temperature range, the Nu of glycol water is lower than that of $CO_2$. The Reynolds number (Re) of glycol water in the low temperature range decreases, which results in a corresponding decrease in the Nu value and flow moves into the laminar flow zone. Due to the fact that glycol water turns into a laminar flow zone at low temperatures, which reduces the heat transfer rate, it is unsuitable as a heat medium for the gasification of cryogenic LNG. Table 2 lists the Prandtl number (Pr), Re, and Nu values of $CO_2$, glycol water, and LNG. Equation (2) defines the convective heat transfer coefficient (h), which influences the overall coefficient of heat transfer (U). Equations (3) and (4) confirm the relationships between Nu and Re and Pr, for the shell and tube, respectively:

$$h = Nu \cdot k/d \tag{2}$$

$$Nu = C\,(Re)^n Pr^{1/2} \tag{3}$$

$$Nu = 0.012\,(Re^{0.87} - 280)Pr^{0.4} \tag{4}$$

where, h is the convective heat transfer coefficient (usually $W/m^2 \cdot K$), Nu is the ratio of convective heat transfer across the boundary, k is the thermal conductivity (usually $W/m \cdot K$), d is the diameter (m) of the tube, C is a constant, Re is the Reynolds number, and Pr is the Prandtl number.

**Table 2.** LNG heat exchanger overall coefficients of heat transfer at optimal conditions. Pr: Prandtl number; Re: Reynolds number.

| | Process | | | |
|---|---|---|---|---|
| | **Shell** | **Tube** | **Shell** | **Tube** |
| **Feed Medium** | Glycol water | LNG | $CO_2$ | LNG |
| **Pr** | 744.6 | 2.1 | 2.5 | 2.1 |
| **Re** | 91.6 | 12,168.9 | 78,714.2 | 123,138.5 |
| **Nu** | 50.81 | 53.32 | 315.39 | 53.19 |

*2.2. Research Area*

Figure 2 presents a schematic diagram of the system used in this study. The vaporizer that transforms LNG into a fuel gas at −163 °C was modeled in three parts based on the LNG state. The temperature limit for the heat medium in front of the $CO_2$ PP (pump) in Figure 2 was set to −32 °C, which is the freezing point for glycol water, and −51 °C, which is the triple point for $CO_2$. The jacket water system was used for all heat sources in the heat medium and the parameters were set to 27 bar and 80 °C. In every heat exchanger, the pressure drops in the shell and tube sides were assumed to be 0.7 and 0.3 bar, respectively. For the pump, we used a conservative assumption of 75% adiabatic efficiency. In this study, the pipe was not taken into consideration.

The properties of the system are listed in Table 3. The system compositions for LNG and glycol water were pure methane and a mixture of water and ethylene glycol, respectively. Additionally, the heat medium used in this study was pure $CO_2$.

**Table 3.** System operating compositions.

| | | **Composition** | **Mole Fraction (%)** |
|---|---|---|---|
| **Feed** | LNG | $CH_4$ | 100 |
| **Heat Medium** | Glycol water | $C_2H_4OH_2$ | 45 |
| | | $H_2O$ | 55 |
| | Pure $CO_2$ | $CO_2$ | 100 |

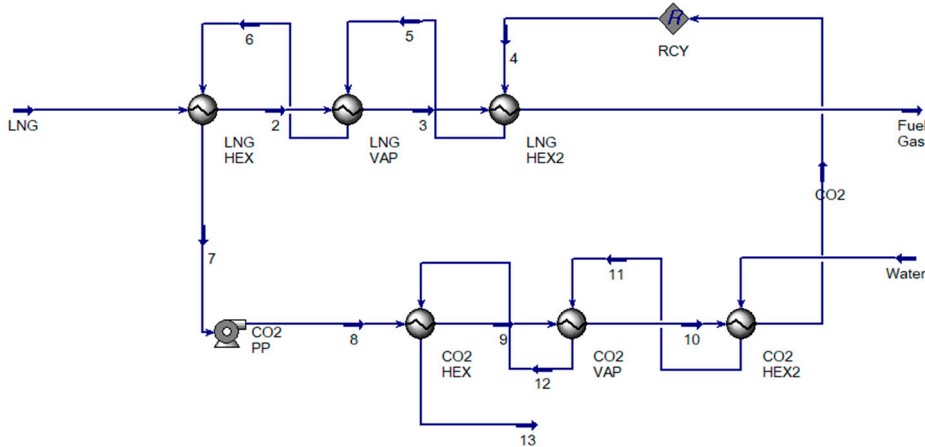

**Figure 2.** Schematic diagram of the $CO_2$ system.

### 2.3. Governing Equations and Optimization

In this study, the Peng–Robinson equation was applied as the equation of state in process modeling using the commercial Aspen HYSYS application [25]. Aspen HYSYS was used for the thermodynamic analysis also as it is known to provide relatively accurate predictions of thermodynamic properties and is widely used in industry as well as the academic field [6,26–28]. The energy balance of each equipment was calculated with its thermodynamic properties under the steady flow condition, as given in Equation (5):

$$\dot{Q} - \dot{W} = \dot{m}(h_{out} - h_{in}) \tag{5}$$

where the changes in the kinetic and potential energies are negligible.

The operating point that minimizes the work at $CO_2$ PP was obtained by adjusting the temperature in front of the LNG HEX2 and the pressure at the back of $CO_2$ PP in Figure 2. The optimal operating point was determined based on the size of the heat exchanger in the system.

The scope of optimization analysis in this study included a pressure range from 18 to 30 bar behind the $CO_2$ PP and a temperature range from 55 to 75 °C in front of the LNG HEX2, as shown in Figure 2. Unlike Bao et al. [21], who used propane to compare the energy recovery based on changes in the condensation temperature, this study detected the optimal operating point that minimizes the work of the pump by adjusting the pressure behind the $CO_2$ PP and the temperature in front of the LNG HEX2 using $CO_2$.

$$SPC = Power_{Pump} / \dot{m}_{LNG} \tag{6}$$

$$U_{Glycol\ Water,CO_2} = \frac{1}{\frac{1}{h_i} + \frac{A_i \ln(r_o/r_i)}{2\pi kL} + \frac{A_i}{A_o}\frac{1}{h_o}} \tag{7}$$

$$A = UA/U = \frac{Q_{LNG}/LMTD_{@Glycol\ Water,CO_2}}{U} \tag{8}$$

The operating point that minimizes the specific power consumption (SPC) was calculated using the power of the pump, which is given as the objective function in Equation (6). At the given operating conditions, we can determine U using Equation (7). The heat exchanger properties were assumed using the shape information of the BEM type in TEMA (Tubular Exchanger Manufactures Association) classification. Type BEM would have the "B" representing the front head, the "E" the core or middle section, and "M" representing the rear head designs.

As Equation (8) shows, the overall heat transfer coefficient multiplies heat transfer area (UA), estimated from process modeling, divided by the value of U estimated from heat exchanger sizing to calculate the heat transfer area A.

## 3. Results

This study compared the specific power consumption (SPC) and changes in the UA and heat transfer area for the two heat media based on the process parameters shown in Figure 2, that is, the temperature in front of the LNG HEX2 and the pressure at the back of the $CO_2$ PP. Furthermore, the SPC and heat transfer area of the heat exchanger were compared at each optimal operating condition for glycol water and $CO_2$. The criteria for system efficiency were determined based on the SPC, as described below.

### 3.1. Specific Power Consumption

Figure 3 presents changes in the SPC as a function of temperature in front of the LNG HEX2 heat source at each operating pressure, where glycol water was used as the heat medium. The change in SPC shows that, as the temperature in front of the LNG HEX2 increases, the SPC decreases without influences from pressure. When the temperature in front of the LNG HEX2 was increased by 20 °C, the performance based on the SPC was improved by 20%.

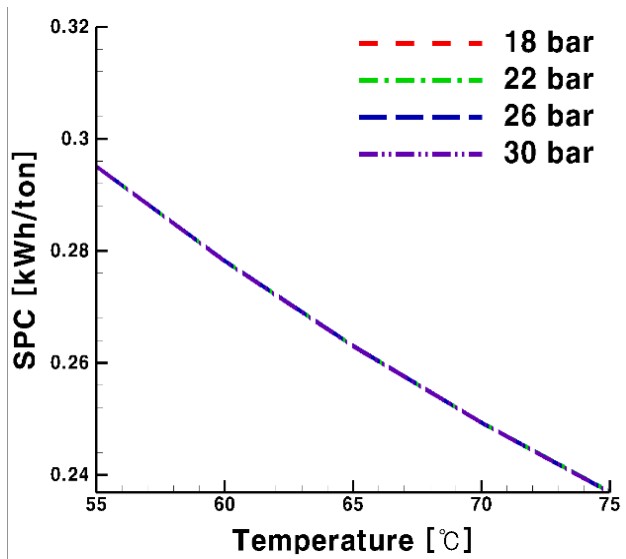

**Figure 3.** Changes in specific power consumption (SPC) as a function of temperature in front of the LNG HEX2 heat source when using glycol water as the heat medium.

Figure 4 presents the changes in SPC as a function of the temperature in front of the LNG HEX2 heat source at each operating pressure, where $CO_2$ was used as the heat medium. The change in SPC shows that, in a similar pattern to Figure 3, an increase in the temperature in front of the LNG HEX2 led to a decrease in the SPC. Furthermore, in contrast to glycol water, $CO_2$ was influenced by pressure behind the $CO_2$ PP. When the inlet temperature of the LNG HEX2 was increased by 20 °C, the performance based on the SPC was improved by 4.2%.

Changes as a function of pressure behind the $CO_2$ PP showed a positive correlation with an increase in pressure and a corresponding increase in the SPC. When the pressure was increased by 12 bar, the performance was dropped by 2%. As shown in Figures 3 and 4, only the temperature in front of the LNG HEX2 influenced the SPC of the glycol water. However, the temperature in front of the LNG HEX2, as well as the pressure at the back of the $CO_2$ PP, influenced the SPC when $CO_2$ was used as the heat medium.

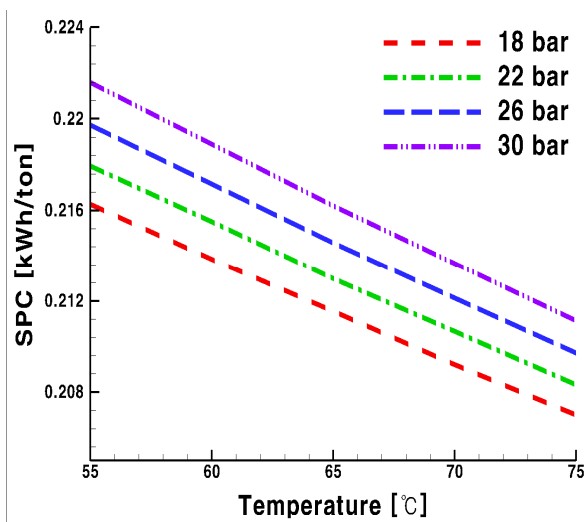

**Figure 4.** Changes in the SPC based on temperature with respect to the operating pressure when using $CO_2$.

### 3.2. Total UA for the Process and Utility Groups with Glycol Water as Heat Medium

Figure 5 presents the changes in UA for the process and utility groups, summation of each three heat exchanger UA, as a function of temperature in front of the LNG HEX2 where glycol water was used as the heat medium. The process group includes the three upper heat exchangers shown in Figure 2 while the utility group includes the three lower heat exchangers. As shown in Figure 5, the UA of the process group decreased with an increase in temperature at the front of the LNG HEX2, whereas there was an increase in the UA of the utility group.

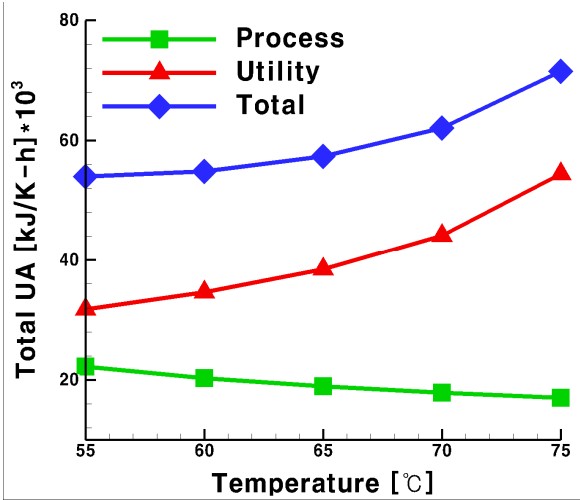

**Figure 5.** Changes in the total heat transfer area (UA) as a function of temperature in each group in front of the LNG HEX2 when using glycol water as the heat medium.

Figure 6 presents a T-Q diagram based on changes in temperature at the front of the LNG HEX2 where glycol water was used as the heat medium. The process in Figure 6 refers to LNG and utility refers to glycol water. The area of the T-Q diagram gives the LMTD. The reason for the change in the UA of the process group in Figure 5 was due to the fact that, as the temperature of the glycol water increased from 55 to 75 °C for LNG gasification, the UA decreased as the LMTD increased based on Equation (1). We were further able to determine that the temperature in front of the LNG HEX2 influenced the UA of the process group, which is the temperature of glycol water, resulting in a decrease in performance by 57% when the temperature increased by 20 °C.

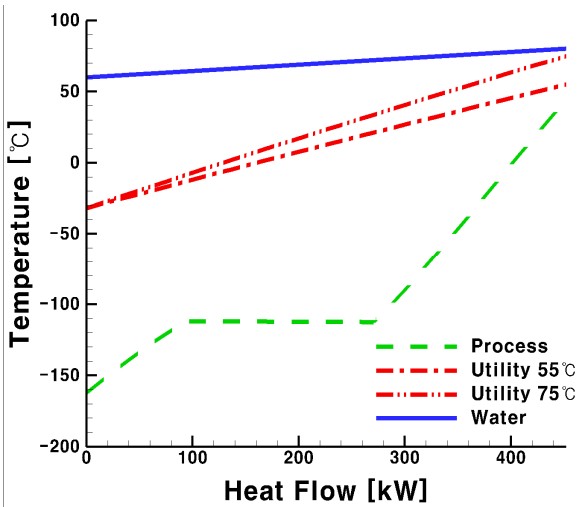

**Figure 6.** Heat transfer performance based on the temperature in each group when using glycol water.

When the temperature of glycol water that exchanges heat with jacket water is increased from 55 to 75 °C, the LMTD, which corresponds to the area of the T-Q diagram, decreased and the UA of the utility group increased. Since the temperature in front of the LNG HEX2 influenced the UA of the utility group, we were further able to determine that performance improved by 21% from that of the UA where the temperature increased by 20 °C.

### 3.3. Total UA for the Process and Utility Groups with $CO_2$ as Heat Medium

Figure 7 shows the total UA of the process and utility groups in front of the LNG HEX2 and behind the $CO_2$ PP where $CO_2$ was used as the heat medium as a function of temperature for specific pressure intervals. We determined that, as the temperature in front of the LNG HEX2 increased, there was a decrease in the UA of the process group and an increase in the UA of the utility group, which was observed at each pressure interval. Furthermore, the total UA decreased as the pressure at the back of the $CO_2$ PP increased.

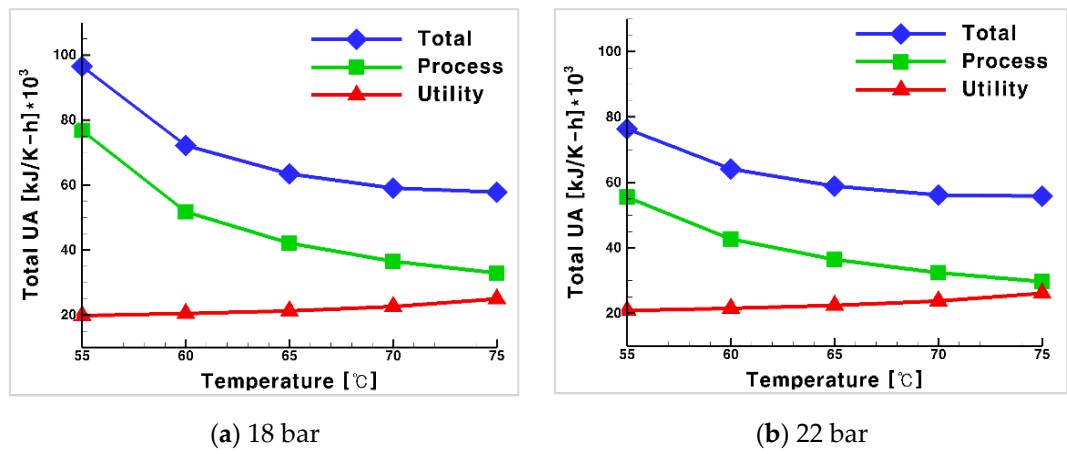

(**a**) 18 bar (**b**) 22 bar

**Figure 7.** *Cont.*

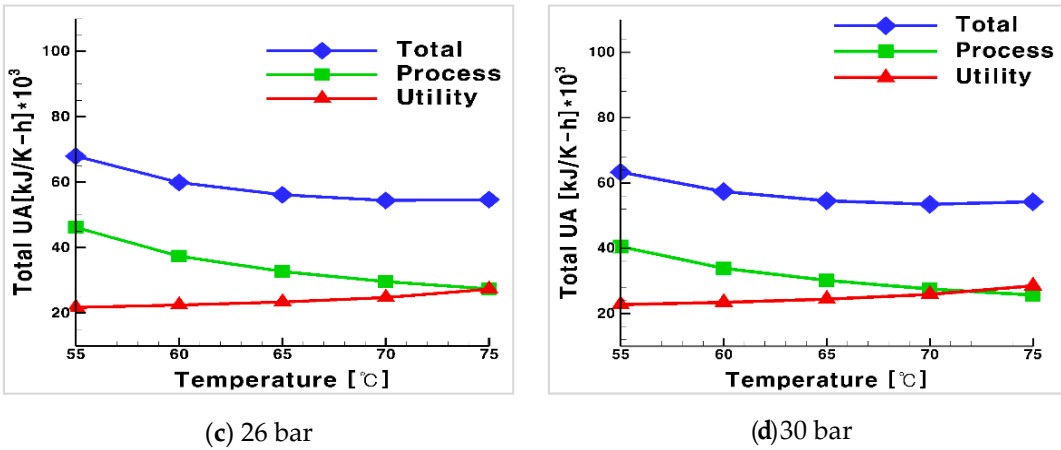

(**c**) 26 bar                  (**d**)30 bar

**Figure 7.** Changes in the total UA based on the pressure in each group when using $CO_2$ (**a**) 18 bar; (**b**) 22 bar; (**c**) 26 bar; (**d**) 30 bar.

### 3.4. Process Optimization Using $CO_2$

The T-Q diagram where $CO_2$ was used as a heat medium shows the changes in the temperature in front of the LNG HEX2 and pressure behind the $CO_2$ PP in Figures 8 and 9, respectively. The LNG is included in the process and $CO_2$ is included in the utility. We observe from these figures that, as the temperature and pressure in the utility increased, the heat exchange area of the process group also increased but the UA decreased with respect to the relationship outlined in Equation (1). We can further observe that both the temperature in front of the LNG HEX2 and pressure behind the $CO_2$ PP affected the UA of the process group. Based on the UA, the performance increased by 37% when the temperature increased by 20 °C and the performance increased by 22% when the pressure increased by 12 bar.

As shown in Figures 8 and 9, as the temperature and pressure increased, the heat exchange area with water decreased while UA increased based on the relationship outlined in Equation (1). Also, as can be seen from Figure 7, the process UA increases. We can further conclude that both the temperature in front of the LNG HEX2 and pressure behind the $CO_2$ PP affect the UA of the utility group. Based on the UA, the performance increases by 4% when the temperature increases by 20 °C and by 20% when the pressure increases by 12 bar.

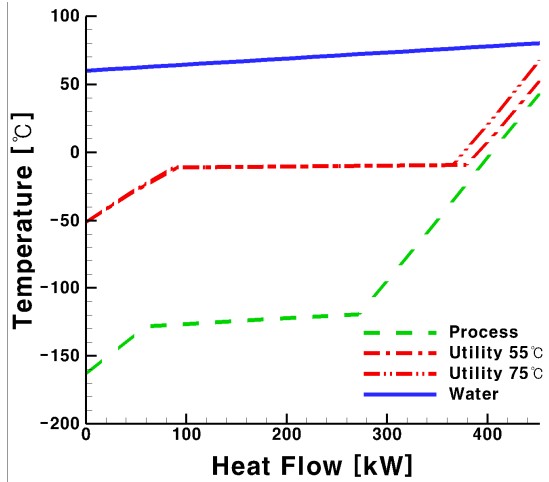

**Figure 8.** Heat transfer performance as a function of temperature in each group when using $CO_2$ as a heat medium.

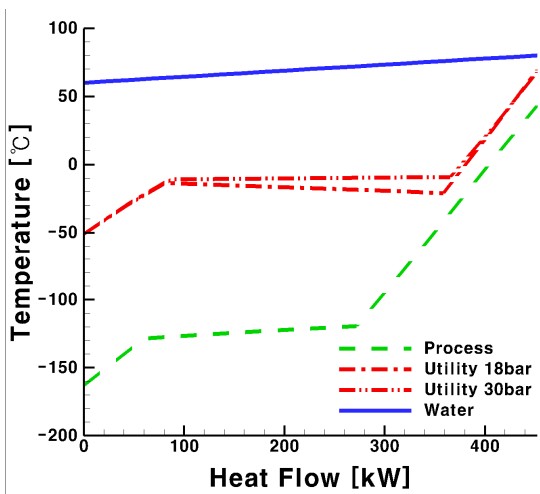

**Figure 9.** Heat transfer performance as a function of pressure in each group when using $CO_2$ as a heat medium.

Table 4 lists the overall heat transfer coefficients for pressure and temperature conditions of 18 bar and 55 °C for glycol water and 30 bar and 75 °C for $CO_2$. Within this system, LNG HEX is a key component and heat exchanger that mediates heat medium exchange with cryogenic LNG. The overall heat transfer coefficient of LNG HEX was determined as approximately half the value when glycol water was used as the heat medium compared with $CO_2$.

**Table 4.** Overall heat transfer coefficients for the heat exchanger at optimal conditions.

| Name | Glycol Water U (kJ/K·h·m²) | $CO_2$ U (kJ/K·h·m²) |
|---|---|---|
| LNG HEX | 117 | 308 |
| LNG VAP | 226 | 280 |
| LNG HEX2 | 191 | 127 |
| HEX | 238 | 452 |
| VAP | - | 414 |
| HEX 2 | - | 191 |

*3.5. Overall Process and Utility Area*

Figures 10 and 11 apply the overall coefficients of heat transfer listed in Table 4, as obtained using Equation (7). When glycol water was used as the heat medium, the heat transfer area increased as the temperature in front of the LNG HEX2 increased, as shown in Figure 10. Conversely, when $CO_2$ was used as the heat medium, the heat transfer area decreased, as shown in Figure 11.

With respect to area, glycol water occupied 28% of the space for a temperature increase of 20 °C and $CO_2$ occupied 23% less space for a temperature increase of 20 °C. It was determined that only the temperature in front of the LNG HEX2 influenced the optimal operating point for the glycol water process efficiency. Thus, lower temperature was responsible for improved efficiency. Conversely, with respect to $CO_2$, the efficiency was improved with an increase in both the temperature in front of the LNG HEX2 and pressure behind the $CO_2$ PP.

Table 5 presents a summary of the results obtained at optimal operation conditions for glycol water and $CO_2$ as the heat medium. The optimal conditions, which were determined to minimize the SPC, are 18 bar and 75 °C for glycol water and $CO_2$, with $CO_2$ characterized by a 14% improvement based on the SPC. The optimal conditions, which were determined to minimize the heat transfer area, are 18 bar and 55 °C for glycol water and 30 bar and 75 °C for $CO_2$. The heat transfer area for the heat exchanger when $CO_2$ was used showed a 7% reduction compared with glycol water. Therefore,

we determined that $CO_2$ is the most suitable heat medium to minimize the size of the heat exchanger in the FGSS.

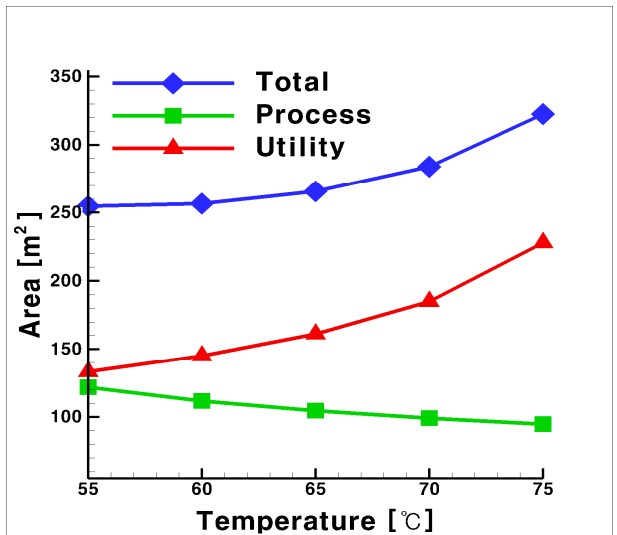

**Figure 10.** Changes in the area as a function of temperature in each group when using glycol water as the heat medium.

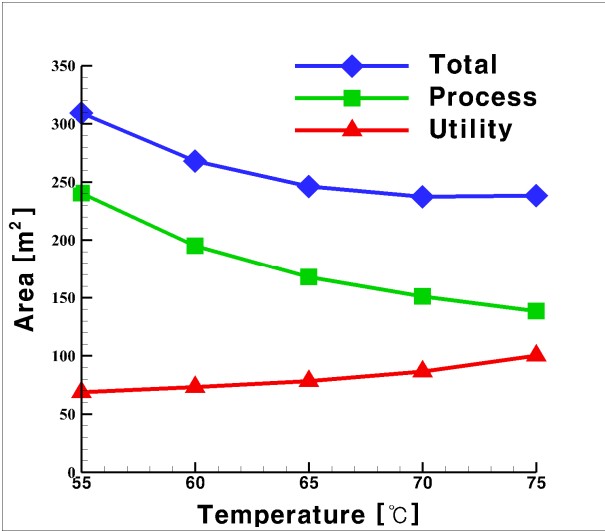

**Figure 11.** Changes in the area as a function of temperature in each group when using $CO_2$ as the heat medium.

**Table 5.** Optimal operating point results.

|  | Glycol Water | Pure $CO_2$ | Ratio (%) |
|---|---|---|---|
| **SPC Operating Point** | 18 bar, 75 °C | | - |
| **SPC (kWh/ton)** | 0.24 | 0.207 | 13.75 |
| **UA Operating Point** | 18 bar, 55 °C | 30 bar, 75 °C | - |
| **UA (kJ/K·h)** | 53,863 | 53,390 | 0.88 |
| **Area Operating Point** | 18 bar, 55 °C | 30 bar, 75 °C | - |
| **Area ($m^2$)** | 255 | 238 | 6.67 |

## 4. Conclusions

An FGS system for an LNG-fueled ship using $CO_2$ as a heat medium was proposed in this study. The system was compared with one using glycol water as a heat medium. The thermodynamic

characteristics were investigated for both systems. System optimization was performed using process modeling while optimizing the system for the temperature and pressure in the heating cycle. The objective functions were divided between the SPC and heat transfer area to define the optimization of the LNG FGSS. The system performance (in terms of SPC) and heat transfer area were compared for the target heat sources (glycol water and $CO_2$). The conclusions of the study are as follows:

(1)  $CO_2$ is the most advantageous heat medium for the gasification of LNG in terms of performance. $CO_2$ enhanced the efficiency (i.e., SPC) by 14% due to the usage of latent heat of vaporization.

(2)  For the system using glycol water, SPC was not affected by pressure change, but lower operating pressure entailed less SPC. For both media, the higher the supply temperature, the lower the required SPC.

(3)  The higher the supply temperature for the system using $CO_2$, the smaller the heat transfer contact area of the process side heat exchanger, but the corresponding value of the utility side heat exchanger was larger. From the viewpoint of the heat transfer area, as the temperature increases, the heat exchanging area decreases.

(4)  Conversely, glycol water exhibited a laminar flow at low temperatures, thereby reducing the heat transfer performance and required a 7% larger heat transfer area for the heat exchanger compared with that using $CO_2$. This result adds support to our conclusion that $CO_2$ is more suitable as a heat medium for the LNG FGS system.

This study also compared the efficiency performance and size of a ship's fuel supply system when replacing the conventional glycol water heat source with $CO_2$. Future studies will focus on a quantitative comparison through experiments as well as analyses of the objective function that encompasses both the SPC and heat transfer area to identify efficiency improvement techniques.

**Author Contributions:** All the authors contributed equally to the paper's preparation.

**Funding:** This work was supported by the Dong-A University research fund.

**Conflicts of Interest:** The authors declare no conflict of interest.

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
