# Peer review of "Comparative Study on Fuel Gas Supply Systems for LNG Bunkering Using Carbon Dioxide and Glycol Water"

_jmse, doi:10.3390/jmse7060184_

Round 1
Reviewer 1 Report
In this paper, the main contribution of this paper is to investigate fuel gas supply system for liquified natural gas (LNG) by using carbon dioxide and glycol water. Moreover, the objective of this study is to minimize the overall device size by increasing the efficiency of the heat transfer process using the latent heat of CO2 vaporization since CO2 is a heat source with a phase change.
As a whole, the research paper is organized properly and also some of the methods are completely described. However, there are some questions need to be answered before making a decision.
1. Lines 10-15, some reasonings can include in the introduction section. The abstract can be revised to include more details of the proposed approach. Authors may improve the abstract by including the existing challenges, motivations, and outcomes of the paper. Abstracts usually have at least one sentence per each: context and background, motivation, hypothesis, methods, results, conclusions.
2. In the Introduction, there is too little literature to support background research. The literature research needs to be completed.
3. Line 143, why don’t authors take into consideration the effect of pipe? What effect does the pipeline have on the system?
4. Authors need to compare and validate the results obtained from the commercial Aspen Hysys application with experimental data and other previously published data from the literature.
5. Line 157, please explain in the manuscript that what the conditions are to allow employing the Peng-Robinson equation.
6. Authors do not provide important experimental data in the abstract and conclusion sections. Therefore, the abstract is insufficient to support the proposed methods and their background.

Reviewer 2 Report
Revision comments are presented in eclosed pdf file.

Reviewer 3 Report
The paper deals with an extremely topical subject in an adequate manner and with a discreet scientific rigor. The quality of the paper is adequate, but my suggestion is to improve the definition of the figures that in some cases have different typeface. Throughout the paper there is a very frequent typing error regarding "CO2" instead of “CO2”, please correct.
Below are some suggestions to improve the final quality of the paper:
Line 35: LNG means Liquefied Natural Gas, so …LNG in liquid state… is to be corrected with …LNG…
Line 53: mistyping “glycol”
Figure 10: correct the typeface for “Temperature” in the graph
Round 2
Reviewer 1 Report
The authors have revised the manuscript considering all the comments carefully. This paper can be a consideration for publication in the present form.
Author Response
Thanks.